# Prognostic and clinicopathological significance of long noncoding RNA MALAT-1 expression in patients with non-small cell lung cancer: A meta-analysis

Xiaoli Liu[1,2], Guichuan Huang[3], Jing Zhang[2], Longju Zhang[3], Zongan Liang[1]*

**1** Department of Respiratory and Critical Care Medicine, West China Hospital of Sichuan University, Chengdu, Sichuan, PR China, **2** Department of Respiratory and Critical Care Medicine, Affiliated Hospital of Zunyi Medical University, Zunyi, Guizhou, PR China, **3** Department of Respiratory and Critical Care Medicine, The First People's Hospital of Zunyi, The Third Affiliated Hospital of Zunyi Medical University, Zunyi, Guizhou, PR China

\* liangza@scu.edu.cn

## Abstract

### Background

Although expression of long non-coding RNA metastasis-associated lung adenocarcinoma transcript 1 (MALAT-1) in tumor tissues has been assessed in several malignancies. However, the association between lncRNA MALAT-1 expression and prognosis or clinicopathological feature remains controversial. Therefore, we conducted a meta-analysis to verify whether lncRNA MALAT-1 expression was associated with prognosis or clinicopathological features in patients with non-small cell lung cancer (NSCLC).

### Methods

We searched Embase, PubMed, Web of Science, Cochrane library, The Chinese National Knowledge Infrastructure, and Wanfang databases from inception to March, 1, 2020. The language restrictions were Chinese and English. The published literature on lncRNA MALAT-I expression and prognosis or clinicopathological characteristics of NSCLC patients was statistically analyzed. Combined hazard ratios (HRs), odds ratios (OR), and 95% confidence intervals (95% CIs) were used to evaluate the effects of lncRNA MALAT-I on the prognosis and clinicopathological features of NSCLC.

### Results

Fifteen studies with 1477 NSCLC patients were enrolled. The results showed that the elevated expression of lncRNA MALAT-I in tumor tissues was associated with shorter overall survival (OS) (HR: 2.20, 95% CI: 1.53–3.16; $P = 0.000$). Additionally, high lncRNA MALAT-I expression was also significantly associated with gender (OR: 0.69, 95% CI: 0.51–0.93; $P = 0.014$), tumor size (OR: 1.87, 95% CI:1.13–3.09; $P = 0.016$), lymph node metastasis (LNM) (OR: 2.87, 95% CI:1.05–7.83, $P = 0.04$), tumor differentiation (OR: 1.60, 95% CI:1.17–2.20; $P = 0.003$), and tumor-node-metastasis (TNM) stage (OR: 0.42, 95% CI: 0.25–0.70; $P =$

**Data Availability Statement:** All relevant data are within the paper and its Supporting Information files.

**Funding:** The authors received no specific funding for this work.

**Competing interests:** NO authors have competing interests.

**Abbreviations:** LncRNA, long non-coding RNA; MALAT-l, metastasis-associated lung adenocarcinoma transcript 1; NSCLC, non-small cell lung cancer; NEAT2, Noncoding nuclear-enriched abundant transcript 2; OS, Overall survival; HRs, Harzard ratios; ORs, Odds rations; Cis, Confidence intervals; qPCR, Quantitative PCR; ISH, In situ hybridization; NOS, Newcastle-Ottawa-Scale; LNM, Lymph node metastasis; LAC, Lung adenocarcinoma.

0.001). There was no significant relationship between lncRNA MALAT-l expression and other clinicopathological features including age (OR: 1.03, 95% CI: 0.79–1.34; *P* = 0.830), number of tumors (OR: 1.02, 95% CI: 0.63–1.64; *P* = 0.943), vascular invasion (OR: 1.23, 95% CI: 0.50–3.05; *P* = 0.652), and recurrence (OR: 1.98, 95% CI: 0.67–5.85; *P* = 0.214).

## Conclusion

The overexpression of lncRNA MALAT-l in NSCLC tissues was correlated with OS, gender, tumor size, LNM, tumor differentiation, and TNM stage. Thus, lncRNA MALAT-l may serve as a prognostic factor for NSCLC.

## Introduction

Cancer is a major public health problem worldwide. According to the statistical analysis of GLOBOCAN 2018, there were approximately 2,093,876 new lung cancer cases and 1,761,007 deaths in 2018 [1, 2]. Radical surgery is the cornerstone for early-stage NSCLC [3] and the 5-year survival rate of these patients exceeds 50% [4]. However, the early symptoms of NSCLC patients are non-specific. More than 60% of patients have middle- or advanced-stage disease at diagnosis and miss the best chance for surgery [5]. Chemotherapy, radiotherapy, and immunotherapy are the preferred treatments for patients with advanced-stage disease [6], but the clinical outcomes are still not encouraging. Therefore, there is an urgent need for a new prognostic factor and therapeutic target for NSCLC.

Metastasis-associated lung adenocarcinoma transcript 1 (MALAT-1) is a long non-coding RNA (lncRNA), also known as nuclear-enriched abundant transcript 2 (NEAT2), which is expressed from chromosome 11q13 and encodes a gene of about 8.7 kb [6]. The lncRNA MALAT-1 is widely expressed in mammalian normal tissues and is abnormally expressed in many human malignancies as well as in NSCLC. Although lncRNA MALAT-1 does not encode a protein, it affects tumor proliferation, apoptosis, drug resistance, invasion, metastasis, and the process of the epithelial-mesenchymal transition, leading to a poor prognosis in patients with malignant tumors. Accumulating evidence indicates that lncRNA MALAT-1 is overexpressed in several types of solid cancers including lung [7], breast [8], gastric [9], bladder [10], and pancreatic [11] cancers. The relationship between MALAT-1 expression and the prognosis of NSCLC remains a subject of debate. Jen *et al.* [12] found that the high expression of MALAT-1 was associated with poor OS and lung tumorigenesis. However, Schmidt *et al.* [13] showed that MALAT-1 expression was not related to prognosis in lung non-squamous cell carcinoma. Through Cox multivariate survival analysis, Mu *et al.* [14] suggested that the decreased expression of MALAT-1 indicated a poor prognosis and was an independent risk factor for NSCLC. It has also been shown that MALAT-1 expression is associated with tumor-node-metastasis (TNM) stage and tumor differentiation [15, 16]; however, these findings remain a subject of debate [17, 18].

This meta-analysis combined published data from studies on MALAT-1 expression and NSCLC prognosis, to determine if MALAT-1 expression has prognostic and clinicopathological significance in patients with NSCLC.

## Material and methods

### Search strategy

The meta-analysis was conducted to compare MALAT-1 expression and the relationship with clinicopathological characteristics and prognosis. We complied with the Preferred Reporting

Items for Systematic Reviews and Meta-analyses (PRISMA) statement [19] to report this study. Embase, PubMed, Web of Science, Cochrane library, The Chinese National Knowledge Infrastructure, and Wanfang databases were used to conduct the meta-analysis. The keywords were as follows: "long noncoding RNA," "lncRNA," "MALAT1," "MALAT-1," "NEAT2," "metastasis-associated lung adenocarcinoma transcript l," "lung cancer," "lung carcinoma," "lung neoplasms," "lung tumor," and "pulmonary cancer," The search terms were connected by the logical conjunctions AND OR according to the Cochrane Handbook, using a combination of free words and subject words, and retrospectively included the references of the study. Titles and abstracts were screened to identify the relevant studies, and then the full texts were read.

### Eligibility criteria

The inclusion criteria for studies were as follows: (1) all patients were diagnosed with NSCLC by pathology; (2) specimens were derived from tumor tissues; (3) studies evaluated the relationship between high MALAT-1 expression and overall survival (OS) or clinicopathological features in patients with NSCLC; (4) MALAT-1 expression in patients with NSCLC was measured by quantitative PCR (qPCR) or *in situ* hybridization (ISH); and (5) published languages were limited to Chinese and English.

The literature exclusion criteria were: (1) case reports, reviews, conference abstracts, and duplicate publications; (2) animal tests; (3) No OS data were available for analysis; and (4) specimens were derived from serum, plasma, or bronchoalveolar lavage fluid.

### Data extraction and quality assessment

Eligible articles were screened independently by two researchers (Xiaoli Liu and Guichuan Huang), according to set criteria. All disagreements at any step of the process were resolved by discussion or by the opinion of a third investigator (Zongan Liang) if necessary. The extracted contents mainly included the following: (1) basic information: first author, country, language, MALAT-1 positive rate; (2) relevant clinical medical record data: sample size, age, gender ratio of each study; (3) pathological characteristics: tumor size, differentiation, lymph node metastasis (LNM), TNM stage, vascular invasion, and recurrence; (4) survival information: hazard ratio (HR) of OS, corresponding to the 95% confidence interval (CI), according to the calculation method of Tierney *et al.* [20]. Cox multivariate regression analysis or Kaplan–Meier survival curve was done without HR or 95% CI. Literature quality was evaluated with the Newcastle-Ottawa Quality Assessment Scale (NOS), using the semi-quantitative principle of the star system to evaluate the quality of the literature. The perfect score is 9 points. A NOS score $\geq$ 6 is classified as a high-quality study.

### Statistical analysis

Stata SE 12.0 software (Stata Corp., College Station, TX, USA) for Windows was used for this meta-analysis, and the HR and its 95% CIs were used to evaluate the relationship between MALAT-1 expression and clinical prognosis in patients with NSCLC. Survival data were obtained from the literature directly or indirectly. Pooled odds ratio (OR) and corresponding 95% CI were used for clinicopathological parameters. The chi-squared test and $I^2$ values were used to assess the heterogeneity among the pooled analysis. When $P > 0.05$ and $I^2 \leq 50\%$, the fixed-effects model was used. By contrast, the random-effects model was used when $P \leq 0.05$ and $I^2 > 50\%$. Subgroup and meta-regression analyses were used for the heterogeneity test. The Begg's test and Egger's test [21, 22] were used to assess publication bias. $P > 0.05$ indicated no potential publication bias. A sensitivity analysis was also applied to evaluate the stability of

the results. If necessary, the trim-and-fill method [23, 24] was used to assess and correct the asymmetry of the funnel graph caused by publication bias. $P < 0.05$ was considered statistically significant.

## Results

### Study identification and selection

Using the outlined search strategy, we identified 2487 citations. Fig 1 shows the study search strategy, according to PRISMA guidelines. After removal of duplicates, the titles and abstracts of 2089 citations were screened according to the inclusion and exclusion criteria. Irrelevant studies were removed, and then the full text of 46 articles was assessed. Of these, 15 papers with 1477 patients met the criteria for further investigation. Table 1 [6, 12–18, 25–31] describes the main characteristics of the selected studies. The studies were published from 2003 to 2020. All studies were case-control studies. Of the 15 studies, 13 were conducted in China and 2 in Germany; a total of 9 studies were published in English and 6 were published in Chinese. The expression of MALAT-1 was detected by qPCR in 14 studies, and by ISH in 1 study. NOS scores were ≥6 of all included studies, and the sample size per literature ranged from 36 to 352.

### Association between lncRNA MALAT-l expression and OS

A total of 10 papers that included 1250 participants reported the information on OS, as shown in Fig 2. A random-effects model was used for the combined HR and 95% CI due to heterogeneity ($P = 0.000$, $I^2 = 73.4\%$). Compared with the MALAT-1 negative (or low) expression group, MALAT-1 positive (or high) expression was associated with a shorter OS in NSCLC, and the difference was statistically significant (HR = 2.20, 95% CI: 1.53–3.16; $P<0.01$). Moreover, significant heterogeneity ($I^2 = 73.4\%$) was observed for OS. We found that the HR and 95% CI values of Mu *et al.* [14] completely opposed those of the other studies, so this study may have been a source of heterogeneity. Therefore, after excluding the Mu study, the other data were recombined (HR = 2.46, 95% CI:1.82–3.34; $I^2 = 58\%$, $P = 0.015$). The results without a change in pooled outcome after removing the Mu study, suggesting that elevated MALAT-1 is correlated with the prognosis of NSCLC. Then we conducted subgroup analysis by publication year, sample size, ethnicity, MALAT-1 assay method, and HR calculation method. As shown in Table 2, MALAT-1 expression was not correlated with subgroup by publication year (≤2016) ($P = 0.149$) and German ethnicity ($P = 0.082$), and there was a significant association with the other groups. Heterogeneity existed in every subgroup, as well as in the multivariate data ($I^2 = 85\%$). Interestingly, there was heterogeneity in the large group (n>70) ($I^2 = 85.2\%$). The heterogeneity of OS was most likely not caused by publication year, sample size, ethnicity, the MALAT-1 assay method, or the HR calculation method. Furthermore, we performed meta-regression analysis by including covariates such as publication year, sample size, ethnicity, MALAT-1 assay method and HR calculation method. Similar to the subgroup analysis, those factors did not result in significant heterogeneity.

### Association between MALAT-1 expression and clinicopathological features

The relationship between the expression of MALAT-1 and the clinicopathological parameters of NSCLC was analyzed from nine aspects: gender, age, tumor size, LNM, tumor differentiation, TNM staging, number of tumors, vascular invasion, and recurrence. As shown in both Fig 3 and Table 3, high MALAT-1 expression was correlated with gender ($P = 0.014$, OR = 0.69, 95% CI: 0.51–0.93, $I^2 = 13.2\%$), Tumor size ($P = 0.016$, OR = 1.87, 95% CI:1.13–3.09, $I^2 = 60.5\%$), LNM ($P = 0.040$, OR = 2.87, 95% CI:1.05–7.83, $I^2 = 83.1\%$), tumor differentiation

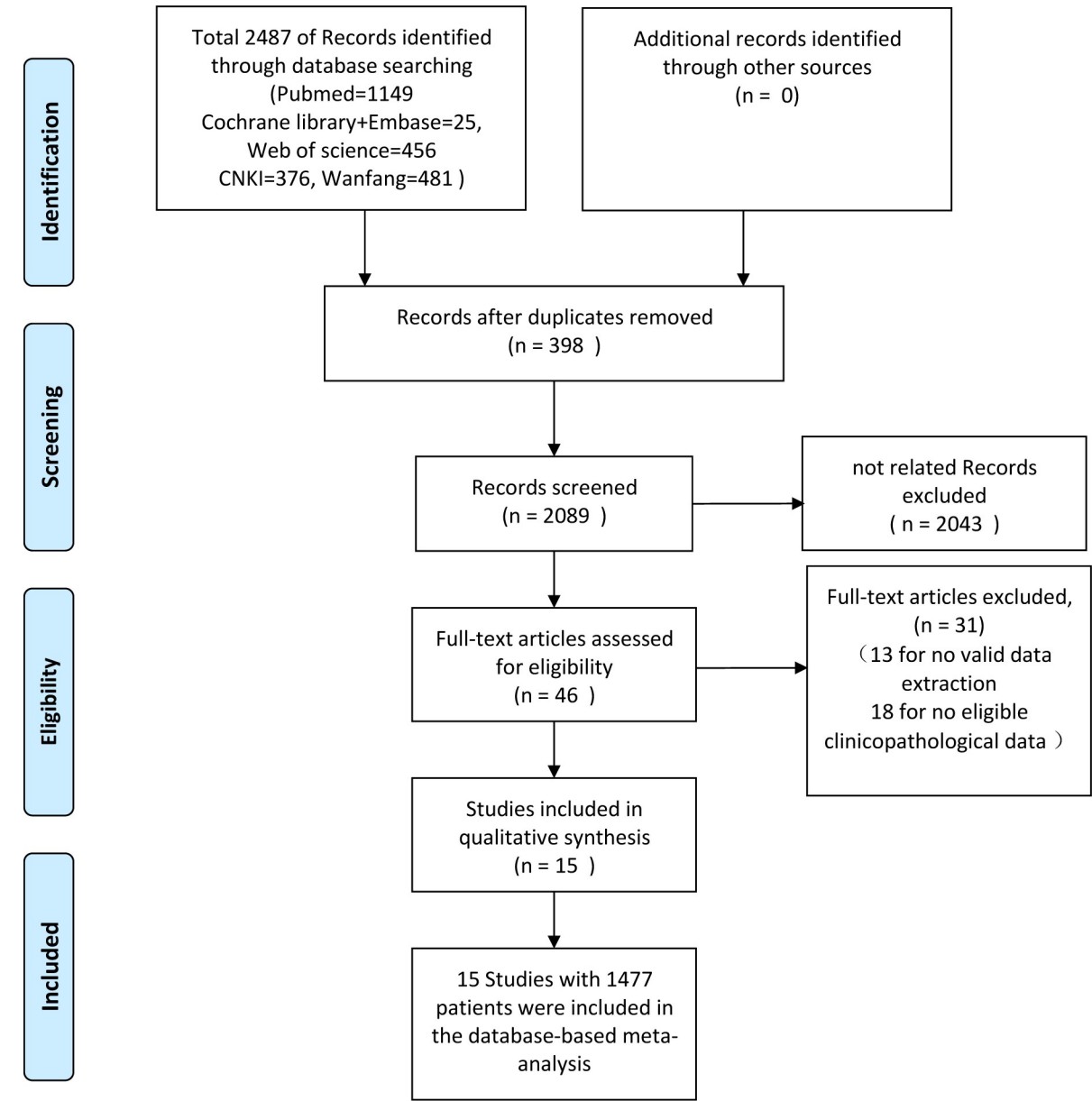

*From:* Moher D, Liberati A, Tetzlaff J, Altman DG, The PRISMA Group (2009). *P*referred *R*eporting *I*tems for *S*ystematic Reviews and *M*eta-*A*nalyses: The PRISMA Statement. PLoS Med 6(7): e1000097. doi:10.1371/journal.pmed1000097

For more information, visit www.prisma-statement.org.

**Fig 1. Flow diagram of the identification and selection of studies.**

**Table 1. Characteristics of the studies included in the meta-analysis.**

| study | country | language | sample size | Detection method | Gender: male(+/-) female(+/-) | age: >50 (+/-) ≤50 (+/-) | tumor size: >3cm(+/-) ≤3cm(+/-) | Differentiation: low (+/-) moderate/high (+/-) | LNM: yes(+/-) no(+/-) | TNM stage: I/II (+/-) III/IV(+/-) | Survial information | NOS scores |
|---|---|---|---|---|---|---|---|---|---|---|---|---|
| Jin, 2020 [25] | China | Chinese | 47 | qPCR | 8/14 13/12 | 8/14 14/11 | 8/17 14/18 | NA | 4/12 17/14 | NA | NA | 6 |
| Yang, 2019 [26] | China | English | 326 | qPCR | 161/66 71/28 | 149/54 83/40 | 197/60 35/34 | 63/14 169/80 | 84/44 148/50 | 66/41 166/53 | OS(M) | 6 |
| Xiao, 2019 [27] | China | English | 39 | qPCR | 10/12 10/7 | 9/5 11/14 | NA | NA | 16/8 4/11 | 9/16 11/3 | OS(S) | 6 |
| Wang, 2018 [28] | China | English | 56 | qPCR | NA | NA | NA | NA | NA | NA | OS(S) | 6 |
| Lin, 2018 [15] | China | English | 120 | qPCR | 40/69 8/3 | 34/37 24/25 | 25/24 33/38 | 32/0 65/23 | NA | 20/32 48/20 | NA | 6 |
| Tang, 2018 [17] | China | English | 36 | qPCR | 7/7 11/11 | 11/7 8/12 | 12/7 4/13 | 12/8 7/9 | 11/9 3/13 | 4/9 12/11 | NA | 6 |
| Peng, 2017 [18] | China | Chinese | 60 | qPCR | 18/20 15/7 | 24/17 8/11 | 16/12 15/17 | 5/7 16/32 | NA | 18/14 15/13 | OS(M) | 7 |
| Chen, 2017 [16] | China | English | 42 | qPCR | 15/12 6/9 | 8/11 13/10 | 14/7 7/14 | 14/13 7/8 | 14/5 7/16 | 6/14 15/7 | OS(M) | 7 |
| Jen, 2017 [12] | China | English | 124 | qPCR | NA | NA | NA | NA | 48/2 59/12 | 60/12 49/2 | OS(S) | 6 |
| Zhang, 2016 [29] | China | Chinese | 125 | qPCR | 44/69 9/3 | 35/44 21/25 | 34/26 32/33 | 24/32 32/37 | NA | 29/24 35/37 | OS(M) | 7 |
| Zhang, 2015 [30] | China | Chinese | 100 | qPCR | 42/19 30/9 | 37/14 35/14 | NA | 43/17 29/11 | NA | NA | NA | 6 |
| Mu, 2013 [14] | China | Chinese | 76 | qPCR | 20/33 2/11 | 2/8 30/36 | 27/40 5/4 | 20/31 12/13 | 15/13 17/31 | 18/29 14/15 | OS(M) | 6 |
| Ma, 2013 [31] | China | Chinese | 86 | qPCR | 47/16 17/6 | 29/23 23/11 | 49/7 17/13 | 16/0 44/26 | 35/1 23/27 | 30/16 39/1 | NA | 6 |
| Schmidt, 2011 [13] | Germany | English | 352 | ISH | NA | NA | NA | NA | NA | NA | OS(S) | 7 |
| Ji, 2003 [6] | Germany | English | 50 | qPCR | NA | NA | NA | NA | NA | NA | OS(S) | 6 |

Abbreviations: qPCR, Quantitative PCR; ISH, in situ hybridization; LNM, lymph node metastasis.NA, not available; OS, overall survival; M, multivariate analysis; S, survival curve; NOS, Newcastle-Ottawa Scale.

($P$ = 0.003, OR = 1.60, 95% CI: 1.17–2.20, $I^2$ = 44.5%), and TNM stage ($P$ = 0.001, OR = 0.42, 95% CI: 0.25–0.70, $I^2$ = 60.6%). However, no significant relationship was observed between MALAT-1 expression and other clinicopathological parameters including age ($P$ = 0.830, OR = 1.03, 95% CI: 0.79–1.34, $I^2$ = 14.9%), number of tumors ($P$ = 0.943, OR = 1.02, 95% CI: 0.63–1.64, $I^2$ = 0.0%), vascular invasion ($P$ = 0.652, OR = 1.23, 95% CI: 0.50–3.05, $I^2$ = 68.3%), and recurrence ($P$ = 0.214, OR = 1.98, 95% CI: 0.67–5.85, $I^2$ = 83.5%). Therefore, the high expression of MALAT-1 in tumor tissue was related to clinicopathological features of NSCLC with regard to the high heterogeneity observed in tumor size ($I^2$ = 60.5%), LNM ($I^2$ = 83.1%), and TNM stage ($I^2$ = 60.6%). Then, we performed subgroup analysis of tumor size, LNM, and TNM stage by publication year and sample size (Table 4). However, MALAT-1 expression was not correlated with LNM in the subgroup of publication year and sample size and significantly heterogeneity. Similarly, meta-regression analysis by publication year and sample size was performed. There was no significant cause of heterogeneity, although heterogeneity in every subgroup existed. The heterogeneity of LNM was not likely caused by publication year and sample

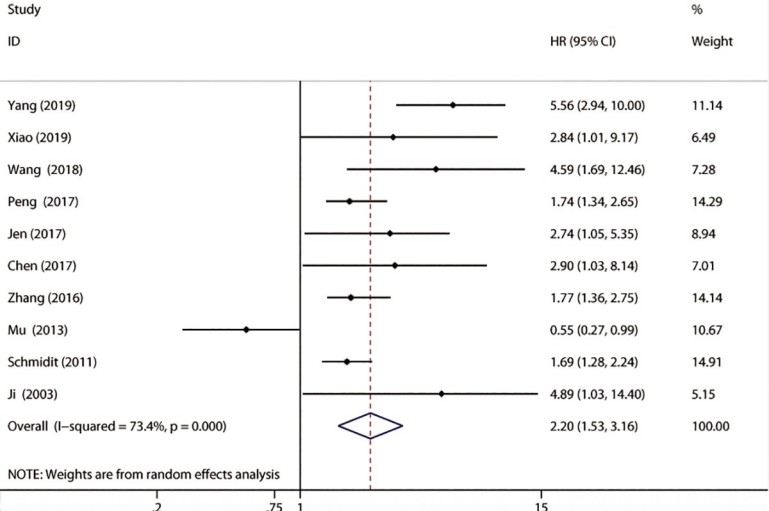

**Fig 2. The combined HR.**

size and cannot be elaborated upon; a large sample of high-quality studies are needed to verify our pooled outcome.

## Publication bias and sensitivity analysis

Ten articles reported the relationship between MALAT-1 expression level and OS, of which five used Cox multivariate analyses, whereas the others used Kaplan–Meier curves. The Begg's test and Egger's test were used to evaluate the publication bias. As shown in Table 5, there was no significant publication bias in the statistical analysis results of OS and other clinicopathological parameters, excluding LNM. Egger's results for LNM suggest publication bias (Egger's

**Table 2. Results of subgroup analysis of OS by publication year, sample size, ethnicity, MALAT-1 assess method, and HR-calculation method.**

| Subgroup analysis | No. of studies | No. Of patients | *P*-value | Pooled HR(95% CI) | Heterogeneity test | | Meta-regression |
|---|---|---|---|---|---|---|---|
| | | | | | $I^2$(%) | *P*-value | (*P* value) |
| **Total** | 10 | 1250 | 0 | 2.20(1.53–3.16) | 73.4 | 0 | |
| **Publication year** | | | | | | | |
| ≤2016 | 4 | 603 | 0.149 | 1.48(0.87,2.51) | 77.9 | 0.004 | 0.096 |
| >2016 | 6 | 647 | 0 | 3.04(1.88,4.89) | 60 | 0.029 | |
| **Sample size** | | | | | | | |
| ≤70 | 5 | 247 | 0 | 2.57(1.64,4.04) | 30.4 | 0.219 | 0.365 |
| >70 | 5 | 1003 | 0.027 | 1.88(1.08,3.29) | 85.2 | 0 | |
| **Ethnicity** | | | | | | | |
| Chinese | 8 | 848 | 0.001 | 2.23(1.40–3.56) | 77.5 | 0 | 0.916 |
| Germany | 2 | 402 | 0.082 | 2.35(0.90–6.14) | 58.1 | 0.122 | |
| **MALAT1** | | | | | | | |
| **Assay Method** | | | | | | | |
| qPCR | 9 | 898 | 0 | 2.36(1.50–3.70) | 75.7 | 0 | 0.634 |
| ISH | 1 | 352 | 0 | 1.69(1.28–2.24) | 0 | 0 | |
| **Analysis type** | | | | | | | |
| Multivariable | 5 | 629 | 0.037 | 1.89(1.04,3.45) | 85 | 0 | 0.396 |
| Unavailable | 5 | 621 | 0 | 2.56(1.62,4.06) | 41 | 0.148 | |

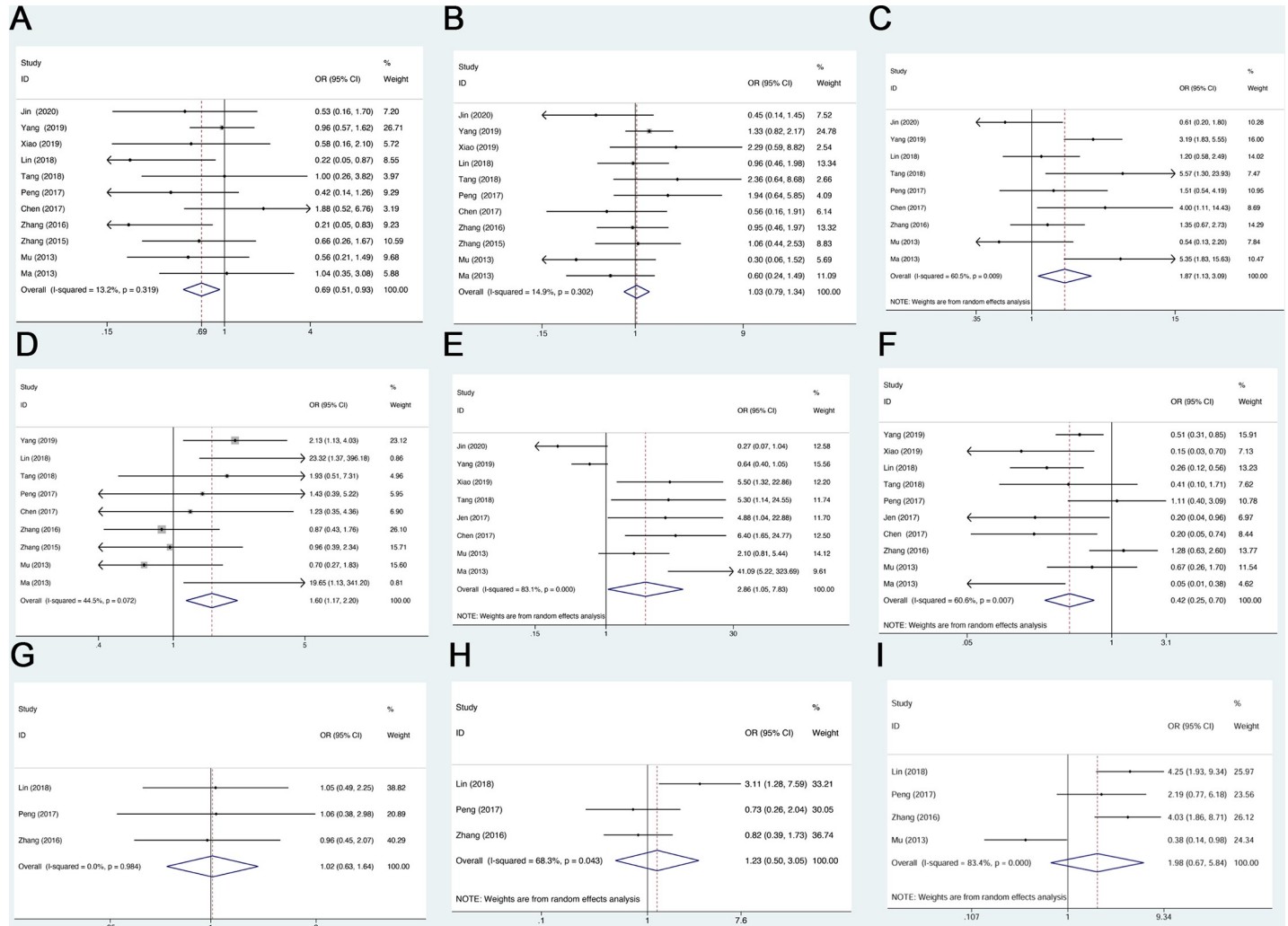

**Fig 3.** Forest plot for the association between MALAT-1 expression and clinicopathological characteristics, including (A) gender. (B) age. (C) tumor size. (D) tumor differentiation. (E) lymph node metastasis. (F) TNM stage. (G) number of tumor. (H) vascular invision. (I) recurrence.

**Table 3. The association between MALAT-1 expression and clinicopathological features.**

| Clinicopathological features | No. of studies | Sample size | p-value | OR(95% CI) | Heterogeneity | | Model |
|---|---|---|---|---|---|---|---|
| | | | | | $I^2$ (%) | p-value | |
| Gender | 11 | 1057 | 0.014 | 0.69(0.51–0.93) | 13.2 | 0.319 | Fixed |
| Age | 11 | 1057 | 0.83 | 1.03(0.79–1.34) | 14.9 | 0.302 | Fixed |
| Tumor size | 9 | 918 | 0.016 | 1.87(1.13–3.09) | 60.5 | 0.009 | Random |
| Differentiation | 9 | 971 | 0.003 | 1.60(1.17–2.20) | 44.5 | 0.072 | Fixed |
| LNM | 8 | 776 | 0.04 | 2.87(1.05–7.83) | 83.1 | 0.000 | Random |
| TNM stage | 10 | 1034 | 0.001 | 0.42(0.25–0.70) | 60.6 | 0.007 | Random |
| No. of tumor | 3 | 305 | 0.943 | 1.02(0.63–1.64) | 0.0 | 0.984 | Fixed |
| Vascular invasion | 3 | 305 | 0.652 | 1.23(0.50–3.05) | 68.3 | 0.043 | Random |
| Recurrence | 4 | 381 | 0.214 | 1.98(0.67–5.85) | 83.5 | 0.000 | Random |

Abbreviation: LNM, lymph node metastasis.

**Table 4. Results of subgroup analysis of clinicopathological features by publication year, sample size and analysis type.**

| Subgroup analysis | No. of studies | No.of patients | *P*-value | Pooled OR(95% CI) | Heterogeneity | | Meta-regression |
|---|---|---|---|---|---|---|---|
| | | | | | $I^2$(%) | P-value | (P *value*) |
| **tumor size** | 9 | 918 | 0.016 | 1.87(1.13,3.09) | 60.5 | 0.009 | |
| Publication year | | | | | | | |
| ≤2016 | 3 | 287 | 0.39 | 1.65(0.53,5.21) | 72.9 | 0.025 | 0.805 |
| >2016 | 6 | 631 | 0.031 | 1.96(1.06–3.61) | 60.1 | 0.028 | |
| Sample size | | | | | | | |
| ≤70 | 4 | 185 | 0.011 | 1.96(0.75,5.14) | 61.2 | 0.052 | 0.92 |
| >70 | 5 | 733 | 0.173 | 1.84(0.96,3.51) | 67.8 | 0.014 | |
| **LNM** | 8 | 776 | 0.04 | 2.87(1.05,7.83) | 83.1 | 0.000 | |
| Publication year | | | | | | | |
| ≤2016 | 2 | 162 | 0.184 | 8.17(0.37,180.83) | 86.6 | 0.006 | 0.379 |
| >2016 | 6 | 614 | 0.182 | 2.16(0.70,6.72) | 82.1 | 0 | |
| Sample size | | | | | | | |
| ≤70 | 4 | 164 | 0.222 | 2.63(0.56,12.43) | 79.4 | 0.002 | 0.859 |
| >70 | 4 | 612 | 0.13 | 3.22(0.71,14.63) | 87.3 | 0 | |
| **TNM stage** | 10 | 1034 | 0.001 | 0.42(0.25,0.70) | 60.6 | 0.007 | |
| Publication year | | | | | | | |
| ≤2016 | 3 | 287 | 0.299 | 0.48(0.12,1.92) | 79.1 | 0.008 | 0.425 |
| >2016 | 7 | 747 | 0.008 | 0.37(0.23,0.61) | 36.2 | 0.152 | |
| Sample size | | | | | | | |
| ≤70 | 4 | 177 | 0.045 | 0.38(0.14,0.98) | 53.4 | 0.092 | 0.86 |
| >70 | 6 | 857 | 0.012 | 0.43(0.22,0.83) | 69.2 | 0.006 | |

test, Pr>|z| = 0.023), and additional Egger's graphs (Fig 4) show crossovers, supporting the conclusions of the Egger's trial. However, the results of Begg's experiment is opposite (Begg's test, Pr>|z| = 0.108), suggesting that there was no publication bias. The reason for the publication bias may be that positive results are easier to publish than negative results. Therefore, we used the trim and fill method to test the publication bias (Fig 5). Due to heterogeneity, outcomes of the random-effects model before and after trimming were (HR: 1.049, 95% CI: 0.056–2.042) and (HR: 0.935, 95% CI: 0.353–2.479). These pooled results did not change before and after trimming, suggesting that the results were stable. Although subgroup analyses of OS and clinicopathological features were conducted by publication year, sample size, and (or) ethnicity and HR calculation method, heterogeneity still existed in every subgroup. We

**Table 5. Publication bias evaluation by Begg's test and Egger's test.**

| Groups of outcomes | No. of studies | Estimates | Begg's test (p-value) | Egger's test (p-value) | Publication bias |
|---|---|---|---|---|---|
| OS | 10 | HR+95%CI | 0.21 | 0.235 | Not significant |
| Gender | 11 | OR+95%CI | 0.213 | 0.153 | Not significant |
| Age | 11 | OR+95%CI | 0.533 | 0.44 | Not significant |
| Tumor size | 9 | OR+95%CI | 0.754 | 0.802 | Not significant |
| Differentiation | 9 | OR+95%CI | 0.076 | 0.125 | Not significant |
| LNM | 8 | OR+95%CI | 0.108 | 0.023 | significant |
| TNM stage | 10 | OR+95%CI | 0.107 | 0.121 | Not significant |
| No. of tumor | 3 | OR+95%CI | 1.000 | 0.702 | Not significant |
| Vascular invasion | 3 | OR+95%CI | 1.000 | 0.918 | Not significant |
| Recurrence | 4 | OR+95%CI | 0.734 | 0.335 | Not significant |

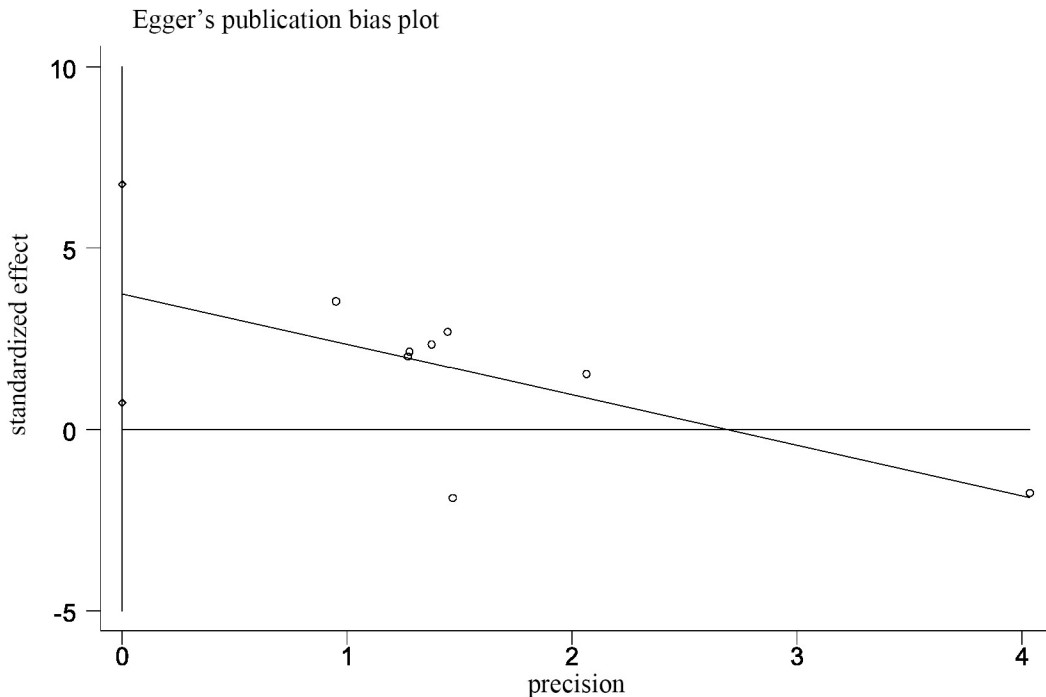

**Fig 4. Egger's bias plot of LNM.**

performed a sensitivity analysis and eliminated single literature to try to find the origin of heterogeneity, As Shown in Figs 6 and 7, the results from a random-effects model suggested that no single paper had a significant effect on the pooled results, and the meta-analyses results were stable.

## Discussion

LncRNAs are recently emerging as key factors of tumorigenesis [32]. LncRNA regulates gene expression at three levels: apparent modification, transcription, and post-transcriptional translation [33]. It is thought to play an important role in the development of many tumors. MALAT-1 is one of the most widely studied lncRNAs. It was first discovered in metastatic human NSCLC, which is believed to promote tumorigenesis. The mechanism of MALAT-1 regulation of tumorigenesis and development may be that MALAT-1 is specifically located at the core of the nucleosomes. This region is involved in aggregation, modification, and/or storage and processing, and thus participates in regulating tumorigenesis [34]. This mapping relationship indicates that MALAT-1 is important in the organization and regulation of gene expression. Therefore, improving the OS rate of patients with NSCLC should be deeply explored from its pathogenesis, to inhibit the proliferation of malignant tumors before invasion and metastasis.

Early studies have shown that aberrant expression of MALAT-1 in metastatic NSCLC [6] plays a pivotal role in carcinogenic development, metastasis, and progression. MALAT-1 has been identified in almost all types of human cancers and is associated with poor patient outcomes [35]. Moreover, Xiao *et al.* [27] found that in human lung adenocarcinoma (LAC) tissues, high level of MALAT1 expression is related to tumor size, TNM stage and LNM, and was

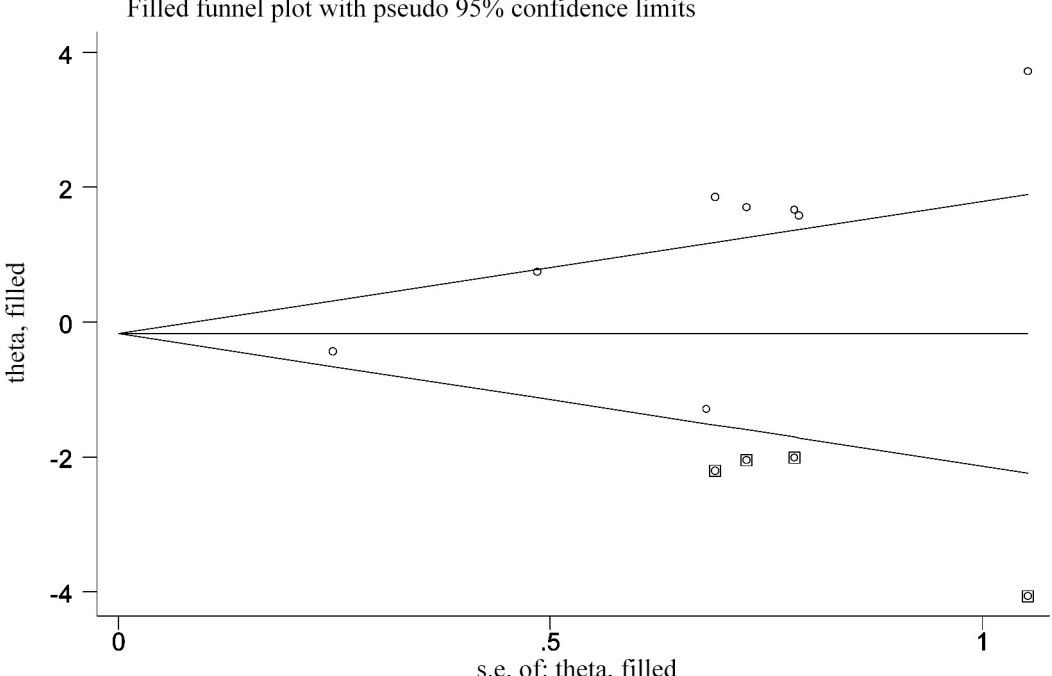

**Fig 5. Trim and fill method of LNM.**

negatively association with miR-429 expression. Wang *et al*. [36] showed that high MALAT1 expression correlated with larger tumor size, lymphatic metastasis, and poorer OS in human gallbladder cancer. Chou *et al*. [37] reported that the aberrant upregulation of MALAT-1 expression correlates with a poor patient prognosis in human breast cancer tissues.

Recently, some meta-analyses have indicated that MALAT-1 has prognostic value in various human cancers as well as NSCLC. The literature included the analysis model; the choice of effect size remains to be discussed. Wu *et al*. [38] and Song *et al*. [39] reported meta-analyses

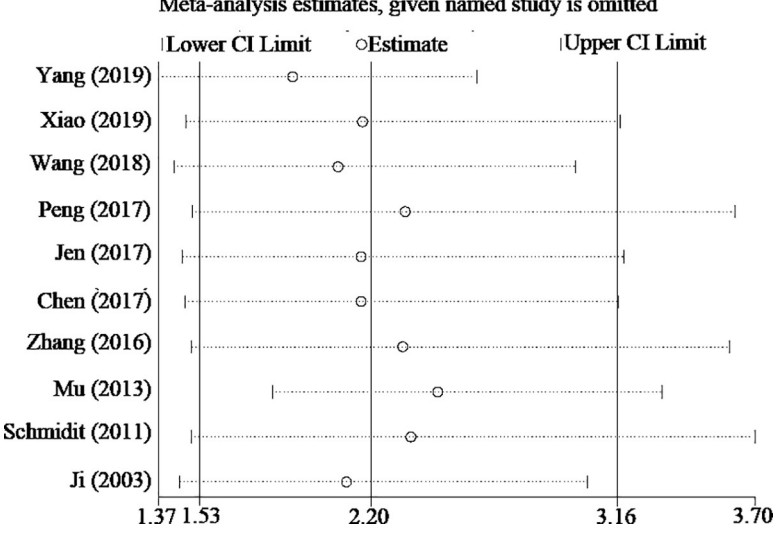

**Fig 6. Sensitivity analysis for meta-analysis of OS.**

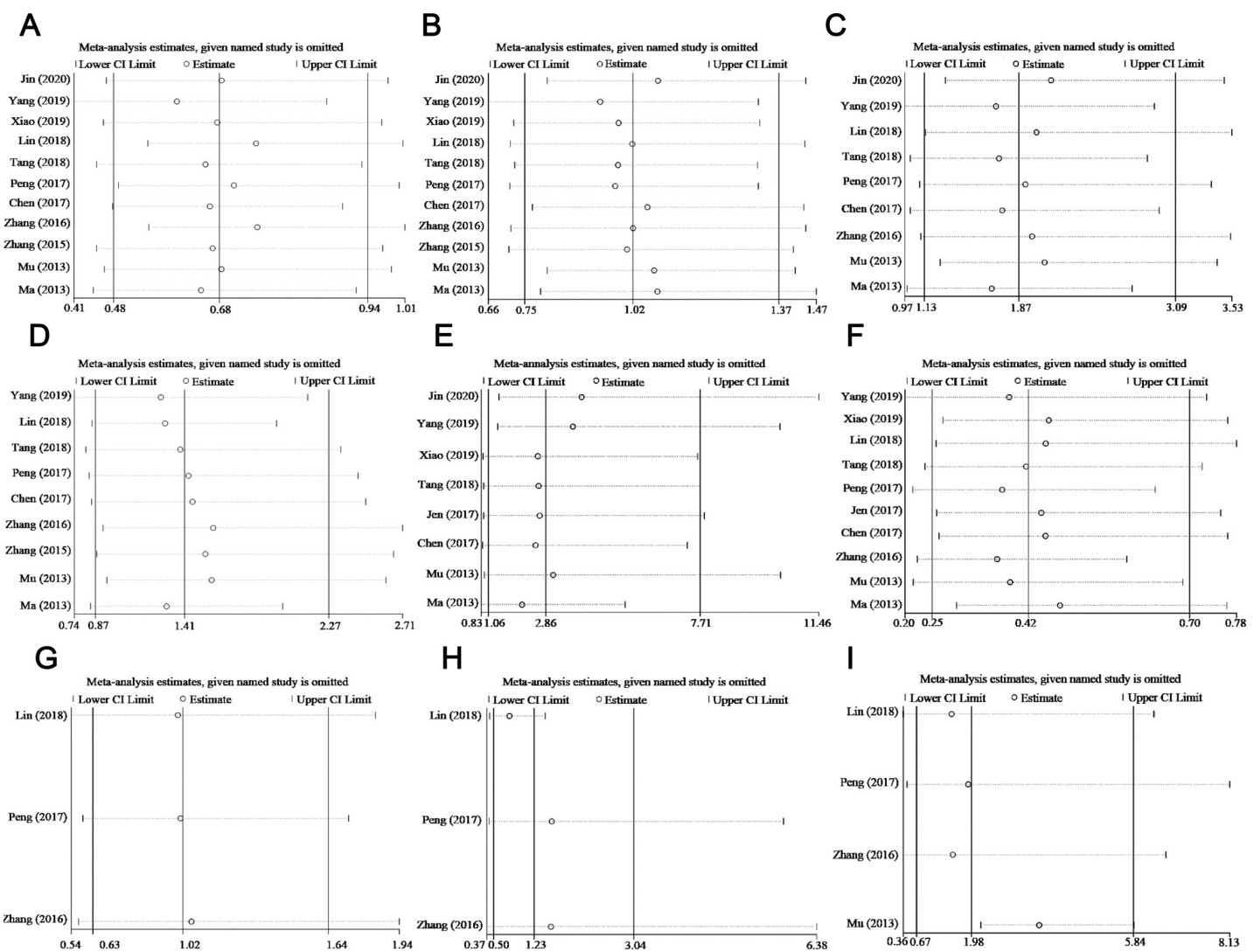

**Fig 7.** Sensitivity analysis for meta-analysis of (A) gender. (B) age. (C) tumor size. (D) tumor differentiation. (E) lymph node metastasis. (F) TNM stage. (G) number of tumor. (H) vascular invision. (I) recurrence.

evaluating the correlation between MALAT-1 expression and the prognosis of cancer patients. In patients with NSCLC, high MALAT-1 expression is associated with a poor OS and DFS. Zhang *et al.* [40] showed that elevated MALAT-1 increased the risk of OS in patients with various human cancers. In subgroup analysis, there was a significant correlation between MALAT-1 and OS in NSCLC. However, Tang *et al.* [41] performed systematic analysis of MALAT-1 and poor prognosis in cancer. Five original studies related to MALAT-1 and NSCLC were included. Subgroup analysis showed that the expression of MALAT-1 was not statistically significant with the prognosis of NSCLC, which was inconsistent with the study by Zhang *et al.* [40]. The fact that only two to five original articles related to MALAT-1 expression and NSCLC were included in the subgroup analysis may have increased the research bias. Other studies in the related meta-analysis showed a statistically significant difference between MALAT-1 expression and NSCLC prognosis. Because it was not a targeted study of NSCLC, only a small number of relevant papers were included.

However, the association between MALAT-1 signification and prognosis or clinicopathological features on the outcome of NSCLC is still controversial. Based on the results of the literature search, we believe this meta-analysis is the first to systematically evaluate the relationship between MALAT-1 expression and clinical significance in NSCLC. Prognostic analysis results from this study showed that high MALAT-1 expression was related to poor OS. Subgroup analysis with a small sample size showed that the heterogeneity significantly decreased ($I^2$ = 30.4%). After removing the Mu *et al.* [14] study, which did not lead to a change in pooled outcome of OS, the results indicated that it may be the main source of heterogeneity in the analysis of the correlation between MALAT-1 expression and OS. The results of relevant clinical pathological parameters analyses showed that MALAT-1 expression related to the gender, tumor size, LNM, tumor differentiation and TNM stage, disclosed that poor tumor differentiation, advanced stage, increased tumor diameter, and LNM are all risk factors for poor prognosis of NSCLC. In NSCLC tissues, patients with high MALAT-1 expression are more likely to develop LNM, large tumor size, tumor differentiation, and TNM stage. These results support the correlation between the positive (high) expression of MALAT-1 and the shorter survival of NSCLC, which can be used as a predictor of NSCLC prognosis.

In this meta-analysis, 10 studies reported that high MALAT-1 expression was correlated with a poor OS, and our pooled results were consistent with these findings. In summary, this study shows that MALAT-1 plays the role of "proto-oncogene" in NSCLC. Additionally, heterogeneity was also found in these combined results. We conducted with meta-regression and sensitivity analyses, and found that the pooled outcomes were credible, no single article influenced the pooled results. Thus, high MALAT-1 expression might be a potential prognosis predicator of NSCLC. The trim and fill method was used to evaluate publication bias and to identify and correct the asymmetry of the funnel graph caused by publication bias, Due to the heterogeneity of the merger, the random-effects model analysis was used. The results of the trimming method were not statistically significant before and after trimming, suggesting that the results were stable. Most of the patients included in these studies were from Asia and there may have been ethnic differences. Studies from different regions are necessary to validate the conclusion.

In the meta-analysis, 3 of 11 studies demonstrated that high MALAT-1 expression was not correlated with gender; however, the pooled outcome indicated that high elevated MALAT-1 expression was significantly correlated with gender. The main cause may be related to different types of histology, and the proportion of patients of different genders is significantly different in our study. Certainly, a lot of high-quality literature is still needed to verify our merged results; 3 of 9 studies considered that positive expression of MALAT-1 was not associated with tumor size, 2 of 10 studies indicated that the expression of MALAT-1 was not related to tumor differentiation, and 1 of 3 studies indicated MALAT-1 expression was correlated with vascular invasion. However, the pooled analysis indicated that high MALAT-1 expression was significantly associated with tumor size, differentiation, LNM and TNM stage, and was not associated with vascular invasion. In the present meta-analysis, the pooled results indicated that MALAT-1 expression was not statistically associated with the numbers of tumor, vascular invasion and recurrence, because of the small number of papers.

## Limitations

There were some limitations in this study. First, all papers included in this study used lung cancer tissue for detection of MALAT-1 expression. Theoretically, the expression of MALAT-1 detected in plasma or serum should be included to predict the prognosis of lung cancer. Blood specimens are easier to obtain in clinical practice and are convenient for long-term monitoring of indicators. However, this study excluded all relevant studies on the expression of MALAT-1 and prognosis of NSCLC in blood samples, for two reasons. One was to reduce

heterogeneity and make the results more stable and clinically convincing. In addition, blood sample detection of MALAT-1 is more for the diagnostic screening of lung cancer. The detection of MALAT-1 in blood specimens needs to be further verified by large and high-quality literature for predicting the prognostic value of lung cancer. Second, part of the HRs and its 95% CIs were extracted from Kaplan-Meier survival curves instead of primary data, which may increase the Heterogeneity; Third, the cut-off value of MALAT-1 expression is different in each study, which may cause heterogeneity. Fourth, tissue sample detect MALAT-1 methods was different may cause further error. Finally, the included studies were conducted in China or Germany; therefore, these conclusions should be treated with caution for other ethnic groups. This paper had publication bias, more high-quality clinical trials and further research on the mechanism of MALAT-1 in NSCLC are needed to provide more convincing evidence for exploring the relationship between MALAT-1 and prognosis of NSCLC.

## Conclusion

Overexpression of MALAT-1 was shown to be significantly correlated with the overall prognosis of NSCLC, and MALAT-1 may serve as a potential prognostic biomarker in NSCLC.

## Supporting information

**S1 Checklist. PRISMA 2009 checklist.**
(DOC)

## Acknowledgments

We would like to thank the authors of the original studies included in this meta-analysis, and the useful comments of the reviewers and the editorial staff.

## Author Contributions

**Data curation:** Jing Zhang, Longju Zhang.

**Supervision:** Zongan Liang.

**Writing – original draft:** Xiaoli Liu.

**Writing – review & editing:** Guichuan Huang.

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
