## [Decision Letter · Decision Letter 0]

22 Jun 2020

PONE-D-20-09605

Prognostic and Clinicopathological Significance of Long Noncoding RNA MALAT-1 Expression in Patients with Non-Small Cell Lung Cancer: A Meta-Analysis

PLOS ONE

Dear Dr. liu,

Thank you for submitting your manuscript to PLOS ONE. After careful consideration, we feel that it has merit but does not fully meet PLOS ONE’s publication criteria as it currently stands. Therefore, we invite you to submit a revised version of the manuscript that addresses the points raised during the review process.

We look forward to receiving your revised manuscript.

Kind regards,

Shama Prasada Kabekkodu

Academic Editor

PLOS ONE

Journal Requirements:

2. At this time, we ask that you please provide the full search strategy and search terms for at least one database used as Supplementary Information.

Reviewers' comments:

Reviewer's Responses to Questions

**Comments to the Author**

1. Is the manuscript technically sound, and do the data support the conclusions?

Reviewer #1: Yes

Reviewer #2: Partly

2. Has the statistical analysis been performed appropriately and rigorously? 

Reviewer #1: Yes

Reviewer #2: Yes

3. Have the authors made all data underlying the findings in their manuscript fully available?

Reviewer #1: Yes

Reviewer #2: Yes

4. Is the manuscript presented in an intelligible fashion and written in standard English?

Reviewer #1: Yes

Reviewer #2: Yes

5. Review Comments to the Author

Reviewer #1: The authors Liang et al., in their article entitled “Prognostic and Clinicopathological Significance of Long Noncoding RNA MALAT-1 Expression in Patients with Non-Small Cell Lung Cancer: A Meta-Analysis” have conducted a meta-analysis to verify the association of lncRNA MALAT1 expression with prognosis and clinicopathological features in patients with non-small cell lung cancer (NSCLC).

The authors have associated an increased GRS with impaired insulin secretion and obesity with high risk for decreased beta-cell function

Review Points

1. The authors have clearly explained the inconsistency in the previous reports and verifying lncRNA MALAT1 expression in lung cancers.

2. The authors have provided detailed explanations for methods and clearly defined inclusion and exclusion criteria.

3. The authors have performed meta-analysis of MALAT1 expression with OS and clinicopathological features sequentially.

4. The authors have also performed sensitivity analysis and tested for publication bias and heterogeneity.

5. The study is satisfactorily executed, but the manuscript contains grammatical errors.

6. The paper requires proofreading and careful editing for grammatical errors.

Following are the clarifications:

1. The authors have stated that “we think publication bias has little effect on the outcome”. The authors need to explain this statement.

2. In the discussion section, the authors have provided information on the results obtained from previous studies. However, it will be helpful for readers if the correlation with the functional impact of MALAT1 expression on features like gender, tumor size, etc. is explained

Reviewer #2: General Overview

The present meta-analysis was performed to investigate the association between MALAT-1 expression with prognosis in NSCLC. For this, a literature search was performed using multiple search engines. Subsequently, statistical analysis for association between MALAT-l and prognosis was undertaken. The meta-analysis includes 15 studies consisting of 1477 NSCLC patients. The findings of the studies were as follows.

1. MALAT-l is overexpressed and associated with shorted OS in NSCLC.

2. High expression of MALAT-l is linked with gender, size of the tumor and LNM and TNM stages.

Based on the analysis, authors propose the prognostic utility of MALAT-l in NSCLC.

Comments

1. More key words should be used.

2. Manuscript should be thoroughly checked for typographical errors, grammar and English language

3. Authors selected studies which had used RT-PCR or ISH as inclusion. Does this also include qPCR?

4. The authors state that “The studies were published from 2003 to 2020”. However, there are no publication in the reference quoting the studies in 2003. Hence, this sentence needs to be modifies.

5. The legends in the Figure S3 , S6 are very small and difficult to read. Need to provide better quality figure with increase fonts.

6. How did the authors grade the study? A GRADE summary of the findings can be given.

7. Already a meta-analysis of MALAT-l in NSCLC is published (PMID: 31846184). Authors should describe, how their meta-analysis does is different from that published in PMID: 31846184. Further, in the limitation of the study, authors mentioned that their meta-analysis is from lung cancer tissue. Why authors did did not include the serological studies in their meta-analysis? A strong and valid justification should be provided. Why authors did did not mention the same in inclusion and exclusion criteria section. Authors can perform reanalysis of the data by including the studies published in PMID: 31846184 and present the data to avoid the possible publication bias.

6. PLOS authors have the option to publish the peer review history of their article (what does this mean?). If published, this will include your full peer review and any attached files.

Reviewer #1: No

Reviewer #2: Yes: Shama Prasada Kabekkodu

---

## [Author Response · Author response to Decision Letter 0]

6 Aug 2020

Journal Requirements:

Question1: Please ensure that your manuscript meets PLOS ONE's style requirements, including those for file naming.

Answer1:

We have revised our manuscript and the charts contained in the manuscript in accordance with the submission guidelines to make it more in line with the publication requirements of PLOS ONE.

Question2：At this time, we ask that you please provide the full search strategy and search terms for at least one database used as Supplementary Information.

Answer 2:

The full search strategy and search terms are as follows:

(((((("Carcinoma, Non-Small-Cell Lung"[Mesh]) OR "Lung Neoplasms"[Mesh]) OR non-small cell lung cancer[Title/Abstract]) OR lung cancer[Title/Abstract])) AND ((((lncRNA[Title/Abstract]) OR long non-coding RNA[Title/Abstract])) OR ((((("MALAT1 long non-coding RNA, human" [Supplementary Concept]) OR NEAT2[Title/Abstract]) OR MALAT1[Title/Abstract]) OR MALAT-1[Title/Abstract]) OR Metastasis associated lung adenocarcinoma transcript 1[Title/Abstract])))

The literature search was from inception of the PubMed database to March 1, 2020, and a total of 1149 papers were retrieved using this search strategy.

Question3：About ORCID

Answer 3:

Thank you for your careful review. We will register an ORCID iD and link it to our Editorial Manager account.

Question4：Answer 4:

This work was not financially supported by any project.

Dear Reviewer #1, 

We thank the reviewer very much for the critical comments and constructive suggestions that have helped to greatly improve our manuscript. We have revised the manuscript according to the suggestions. Our responses to the comments are as follows:

Reviewer #1: 

The authors Liang et al., in their article entitled “Prognostic and Clinicopathological Significance of Long Noncoding RNA MALAT-1 Expression in Patients with Non-Small Cell Lung Cancer: A Meta-Analysis” have conducted a meta-analysis to verify the association of lncRNA MALAT1 expression with prognosis and clinicopathological features in patients with non-small cell lung cancer (NSCLC).

Question1. The authors have stated that “we think publication bias has little effect on the outcome”. The authors need to explain this statement.

Answer 1:

According to your suggestion, we have discussed this possibility in our manuscript. Indeed, statements such as “we think publication bias has little effect on the outcome” are not rigorous and require further explanation and revision. The trim and fill method [1] was used to test publication bias and to identify and correct the asymmetry of the funnel graph caused by publication bias. We used Egger’s and Begg’s methods to test the publication bias. When we used Egger’s method to evaluate the LNM group data, we found that there was publication bias (P=0.023), so we used the trim and fill method [2, 3] to verify the publication bias. Because of the data heterogeneity, we used the random-effects model to analyze the results of the trim and fill method. The pooled results before and after trimming did not change [before trimming: (HR:1.049, 95% CI: 0.056–2.042), after trimming: (HR: 0.935 95% CI: 0.353–2.479)], so we believe that the results were stable, and the merged data were credible. （page24, 398-404）

References

[1]. Zhang T.S, Zhong W.Z. [Performance of the Nonparametric Trim and Fill Method in Stata]. [J] Evidence-Based Medicine. 2009;9(04):240-2.(Chinese)

[2]. Duval S, Tweedie R. Trim and fill: A simple funnel-plot-based method of testing and adjusting for publication bias in meta-analysis. Biometrics. 2000;56(2):455-63. doi: 10.1111/j.0006-341x.2000.00455.x. PubMed PMID: 10877304.

[3]. Peters JL, Sutton AJ, Jones DR, Abrams KR, Rushton L. Performance of the trim and fill method in the presence of publication bias and between-study heterogeneity. Stat Med. 2007;26(25):4544-62. Epub 2007/05/04. doi: 10.1002/sim.2889. PubMed PMID: 17476644.

Question2. In the discussion section, the authors have provided information on the results obtained from previous studies. However, it will be helpful for readers if the correlation with the functional impact of MALAT1 expression on features like gender, tumor size, etc. is explained

Answer 2:

We have made revised the manuscript based on your suggestion; the revisions are in red font. (page23- 24, lines 382–390；page 25, lines 408-415)

Dear Dr. Shama Prasada Kabekkodu, 

Thank you very much for your comments on our manuscript entitled, “Prognostic and Clinicopathological Significance of Long Noncoding RNA MALAT-1 Expression in Patients with Non-Small Cell Lung Cancer: A Meta-Analysis” (PONE-D-20-09605). We believe that your comments will make our manuscript more acceptable. 

Reviewer #2

General Overview

The present meta-analysis was performed to investigate the association between MALAT-1 expression with prognosis in NSCLC. For this, a literature search was performed using multiple search engines. Subsequently, 

Comments：

1. More key words should be used.

2. Manuscript should be thoroughly checked for typographical errors, grammar and English language

4. The authors state that “The studies were published from 2003 to 2020”. However, there are no publication in the reference quoting the studies in 2003. Hence, this sentence needs to be modifies.

5. The legends in the Figure S3 , S6 are very small and difficult to read. Need to provide better quality figure with increase fonts.

Answer 1, 2, 5:

We have corrected these mistakes in the revised manuscript and added useful keywords. We have also carefully checked the grammar and spelling, and re-edited the picture to make it easier to read.

Answer 4:

The study of 2003 reference cited is listed below:

[6] Ji P, Diederichs S, Wang W, Boing S, Metzger R, Schneider PM, et al. MALAT-1, a novel noncoding RNA, and thymosin beta4 predict metastasis and survival in early-stage non-small cell lung cancer. Oncogene. 2003;22(39):8031-41. Epub 2003/09/13. doi: 10.1038/sj.onc.1206928. PubMed PMID: 12970751.

Question3.Authors selected studies which had used RT-PCR or ISH as inclusion. Does this also include qPCR?

Answer 3:

After careful reading and inspection of the original literature, we found that all selected studies (except Schmidt et al 2011) used quantitative PCR to detect MALAT-1 expression, so we made corrections to the revised manuscript and to Table 1.

Question6: How did the authors grade the study? A GRADE summary of the findings can be given.

Answer 6:

Thank you very much for your valuable comments. To evaluate the quality of the literature, we re-evaluated the original literature and summarized the NOS into a separate table at the end of the letter, and also revised it in Table 1. 

Question7: Already a meta-analysis of MALAT-l in NSCLC is published (PMID: 31846184). Authors should describe, how their meta-analysis does is different from that published in PMID: 31846184. Further, in the limitation of the study, authors mentioned that their meta-analysis is from lung cancer tissue. Why authors did did not include the serological studies in their meta-analysis? A strong and valid justification should be provided. Why authors did not mention the same in inclusion and exclusion criteria section. Authors can perform reanalysis of the data by including the studies published in PMID: 31846184 and present the data to avoid the possible publication bias.

Answer 7:

Thank you very much for your pertinent comments. We found PMID:31846184 article for Zheng et al. “Long noncoding RNA MALAT1 as a candidate serological biomarker for the diagnosis of non-small cell lung cancer: A meta-analysis.” Comparing the differences between the two articles, we think that study（PMID: 31846184）entitled, “Diagnostic studies relevant to circulation long noncoding RNA MALAT1 as a candidate serological biomarker for NSCLC,” which used meta-analysis of diagnostic tests, including six studies with eight datasets, analysis of circulating blood lncRNA MALAT-1 expression sensitivity, specificity, positive likelihood ratio, negative likelihood ratio and diagnostic odds ratio, suggest that it has a high diagnostic value for NSCLC and a low misdiagnosis rate. However, our study takes into account the correlation of high expression of lncRNA MALAT-1 and the prognosis of patients with NSCLC. It also used meta-analysis of survival data to analyze the relationship between the expression of lncRNA MALAT-1 and the prognostic survival of patients with NSCLC. We think the focus of these article are different, serum long noncoding RNA MALAT1 is a promising biomarker for NSCLC screening. In our study, tissue samples for MALAT-1 were mainly aimed at the prognosis assessment of NSCLC. In addition, in order to reduce statistical heterogeneity and make the results more stable, credible, and more convincing, we only included lung tissue specimens as the research object, tissue sample can make the meta-analysis results more Reliable and more clinically convincing. At the same time, in the suggestions you give us, we also found that the exclusion criteria and the Discussion section were not rigorous. We have revised the manuscript and marked the revisions in red font (page 7, lines 127–128; page 26, lines 431-440).

Table Methodological Quality of Studies Included in the Final Analysis Based on the Newcastle-Ottawa-Scale for Assessing the Quality of Case-Control Studies

Studies Selection(score) Comparability(score) Exposure(score) Total scores

 the case definition adequate Representativeness

 of the cases Selection of Controls Definition of Controls Basis on the

 design or analysis Ascertainment 

of exposure Same methods of ascertainment Non-Response rate 

Jin et al.(2020)25 1 1 0 0 1 1 1 1 6

Peng et al.(2017)18 1 1 0 0 2 1 1 1 7

Zhang et al.(2016)29 1 1 0 0 2 1 1 1 7

Zhang et al.(2015)30 1 1 0 0 1 1 1 1 6

Mu et al.(2013)14 1 1 0 0 1 1 1 1 6

Ma et al.(2013)31 1 1 0 0 1 1 1 1 6

Yang et al.(2019)26 1 1 0 0 1 1 1 1 6

Xiao et al.(2019)27 1 1 0 0 1 1 1 1 6

Lin et al.(2018)15 1 1 0 0 1 1 1 1 6

Tang et al.(2018)17 1 1 0 0 1 1 1 1 6

Jen et al.(2017)12 1 1 0 0 1 1 1 1 6

Schmidt et al.(2011)13 1 1 0 0 2 1 1 1 7

Ji et al.(2003)6 1 1 0 0 1 1 1 1 6

Wang et al.(2018)28 1 1 0 0 1 1 1 1 6

Chen et al.(2017)16 1 1 0 0 2 1 1 1 7

---

## [Decision Letter · Decision Letter 1]

24 Sep 2020

Prognostic and Clinicopathological Significance of Long Noncoding RNA MALAT-1 Expression in Patients with Non-Small Cell Lung Cancer: A Meta-Analysis

PONE-D-20-09605R1

Dear Dr. Liang,

We’re pleased to inform you that your manuscript has been judged scientifically suitable for publication and will be formally accepted for publication once it meets all outstanding technical requirements.

Kind regards,

Shama Prasada Kabekkodu

Academic Editor

PLOS ONE

Additional Editor Comments (optional):

Both the reviewers have recommended the publication of our your manuscript.

Reviewers' comments:

Reviewer's Responses to Questions

**Comments to the Author**

1. If the authors have adequately addressed your comments raised in a previous round of review and you feel that this manuscript is now acceptable for publication, you may indicate that here to bypass the “Comments to the Author” section, enter your conflict of interest statement in the “Confidential to Editor” section, and submit your "Accept" recommendation.

Reviewer #1: (No Response)

Reviewer #2: All comments have been addressed

2. Is the manuscript technically sound, and do the data support the conclusions?

Reviewer #1: (No Response)

Reviewer #2: Yes

3. Has the statistical analysis been performed appropriately and rigorously? 

Reviewer #1: (No Response)

Reviewer #2: Yes

4. Have the authors made all data underlying the findings in their manuscript fully available?

Reviewer #1: (No Response)

Reviewer #2: Yes

5. Is the manuscript presented in an intelligible fashion and written in standard English?

Reviewer #1: (No Response)

Reviewer #2: Yes

6. Review Comments to the Author

Reviewer #1: (No Response)

Reviewer #2: Authors of the manuscript "Prognostic and Clinicopathological Significance of Long Noncoding RNA MALAT-1 Expression in Patients with Non-Small Cell Lung Cancer: A Meta-Analysis " have satisfactorily addressed all my concerns.

7. PLOS authors have the option to publish the peer review history of their article (what does this mean?). If published, this will include your full peer review and any attached files.

Reviewer #1: No

Reviewer #2: **Yes: **Shama Prasada Kabekkodu

---

## [Editor Report · Acceptance letter]

1 Oct 2020

PONE-D-20-09605R1 

Prognostic and Clinicopathological Significance of Long Noncoding RNA MALAT-1 Expression in Patients with Non-Small Cell Lung Cancer: A Meta-Analysis 

Dear Dr. Liang:

I'm pleased to inform you that your manuscript has been deemed suitable for publication in PLOS ONE. Congratulations! Your manuscript is now with our production department. 

Kind regards, 

on behalf of

Dr. Shama Prasada Kabekkodu 

Academic Editor

PLOS ONE